# PER2 Circadian Oscillation Sensitizes Esophageal Cancer Cells to Chemotherapy

**DOI:** 10.3390/biology10040266

**Published:** 2021-03-26

**Authors:** Juan Alfonso Redondo, Romain Bibes, Alizée Vercauteren Drubbel, Benjamin Dassy, Xavier Bisteau, Eleonore Maury, Benjamin Beck

**Affiliations:** 1Institute of Interdisciplinary Research (IRIBHM), Faculty of Medicine, Erasme Campus of Université Libre de Bruxelles (ULB), 808 Route de Lennik, 1070 Brussels, Belgium; jredondo@ulb.be (J.A.R.); rbperso@e.email (R.B.); Alizee.Vercauteren.Drubbel@ulb.be (A.V.D.); Benjamin.Dassy@ulb.be (B.D.); Xavier.Bisteau@ulb.be (X.B.); 2Endocrinology, Diabetes and Nutrition Unit, Institute of Experimental and Clinical Research (IREC), Faculty of Medicine, Bruxelles-Woluwe Campus of Université catholique de Louvain (UCLouvain), 55 Avenue Hippocrate, 1200 Woluwe-Saint-Lambert, Belgium; eleonore.maury@uclouvain.be

**Keywords:** circadian clock, esophagus cancer, squamous cell carcinoma, apoptosis, chemotherapy, chronotherapy

## Abstract

**Simple Summary:**

There are growing evidences that the circadian rhythm modulates key cellular processes in physiological and pathological conditions. Here, we characterized the consequences of the daily oscillations of the clock-related gene PER2 in esophageal cancer cells and found that chemotherapy is more efficient when PER2 expression is low. These results suggest that chronotherapy might be used to potentiate the impact of current chemotherapy regimen.

**Abstract:**

Esophageal squamous cell carcinoma (eSCC) accounts for more than 85% cases of esophageal cancer worldwide and the 5-year survival rate associated with metastatic eSCC is poor. This low survival rate is the consequence of a complex mechanism of resistance to therapy and tumor relapse. To effectively reduce the mortality rate of this disease, we need to better understand the molecular mechanisms underlying the development of resistance to therapy and translate that knowledge into novel approaches for cancer treatment. The circadian clock orchestrates several physiological processes through the establishment and synchronization of circadian rhythms. Since cancer cells need to fuel rapid proliferation and increased metabolic demands, the escape from circadian rhythm is relevant in tumorigenesis. Although clock related genes may be globally repressed in human eSCC samples, PER2 expression still oscillates in some human eSCC cell lines. However, the consequences of this circadian rhythm are still unclear. In the present study, we confirm that PER2 oscillations still occur in human cancer cells in vitro in spite of a deregulated circadian clock gene expression. Profiling of eSCC cells by RNAseq reveals that when PER2 expression is low, several transcripts related to apoptosis are upregulated. Consistently, treating eSCC cells with cisplatin when PER2 expression is low enhances DNA damage and leads to a higher apoptosis rate. Interestingly, this process is conserved in a mouse model of chemically-induced eSCC ex vivo. These results therefore suggest that response to therapy might be enhanced in esophageal cancers using chronotherapy.

## 1. Introduction

In complex multicellular organisms, the circadian clock orchestrates several physiological processes and behaviors through the establishment and synchronization of circadian rhythms [1,2]. The circadian clock is a cell autonomous process. It is composed of multiple transcription regulators that participate in interconnected transcriptional feedback loops to generate their own oscillation. The clock is driven by positive regulators CLOCK and BMAL1, which regulate expression of Cryptochromes (CRY1 and CRY2) and Period (PER1, PER2, and PER3) [3]. Circadian clock components directly or indirectly regulate the expression of hundreds of genes in different cell types. This leads to daily rhythms in many cellular processes including nutrient metabolism, redox regulation, autophagy, DNA damage repair, protein folding, and cellular secretion [4]. It has been shown that nearly half of the mouse genome oscillates with circadian rhythm, highlighting the pivotal role of the circadian clock in cell biology [5]. Since daily rhythm in numerous cellular processes is a crucial part of homeostasis, it is not surprising that perturbation of the clock is associated to pathological conditions such as neurodegeneration [6], metabolic disorders [7], and cancer [8,9].

Since cancer cells need to fuel rapid proliferation and increased metabolic demands, the escape from circadian rhythm is relevant in tumorigenesis. Studies have shown that cancer cells commonly display a disrupted circadian rhythm or a deregulated expression of the circadian clock genes [10]. In the same line of thought, loss of circadian clock gene expression has been correlated with poor prognosis in various cancer types such as breast cancer [11], head and neck cancer [12], and gastric cancer [13]. This loss of the molecular clock may even participate in resistance to apoptosis [14,15] or immune evasion [2]. Nonetheless, this does not mean that circadian clock related genes do not play a role in cancer cell physiology. Indeed, other pieces of evidence have revealed that dysfunctions in circadian rhythms affect tumorigenesis and clock genes regulate several cancer hallmarks [16]. It still remains unclear in which cancers circadian rhythms are altered and whether it may be exploited to improve current therapeutic approaches. Notably, very little is known about the circadian clock in esophageal cancers.

Esophageal cancer is the eighth most frequent cancer in the world and is associated to a poor survival rate due its aggressive nature. Esophageal carcinoma occurs as either squamous cell carcinoma (SCC) or adenocarcinoma. Esophageal SCC (eSCC) accounts for more than 85% cases of esophageal cancer worldwide [17] and the 5-year survival rate associated with metastatic esophageal cancer is poor (5% in the United States) [18]. This low survival rate is the consequence of a complex mechanism of resistance to therapy and tumor relapse. To effectively reduce the mortality rate in these patients, we need to better understand the molecular mechanisms underlying the development of this deadly disease and translate that knowledge into novel approaches for its early diagnosis, treatment, and prevention. Very little is known about the circadian clock in esophageal cancers. Recent studies have suggested that clock related genes might be globally repressed in human eSCC samples as compared to normal tissue [19]. In spite of these alterations, PER2 oscillations still occur in some esophageal cancer cell lines [19]. Nonetheless, the consequences of these oscillations are still unclear.

In the present study, we show that clock related gene expression is altered in esophageal SCC cells. By using a bioluminescent reporter under the control of the PER2 promoter in esophageal cells, we have observed the oscillation of PER2 expression in human cancer cells in vitro in spite of the deregulated circadian clock gene expression. Characterization of esophageal cancer cells by RNAseq indicated that when PER2 expression is low, several transcripts related to the apoptotic signaling pathway are upregulated. To determine whether this oscillation would modify the response to therapy, we have treated eSCC cells with cisplatin. Strikingly, treating esophageal cancer cells with cisplatin for 4 h when PER2 expression is high or low has different outcomes. Indeed, eSCC cells undergo apoptosis more frequently when PER2 expression is low. Interestingly, this process is conserved in a mouse model of chemically-induced eSCC. These results therefore suggest that the response to therapy might be enhanced in esophageal cancers using chronotherapy.

## 2. Materials and Methods

### 2.1. The Cancer Genome Atlas Analysis

To obtain public Esophageal Carcinoma RNA sequencing data and the corresponding clinical information The Cancer Genome Atlas program (TCGA) was used. Illuminahiseq_rnaseqv2-RSEM_genes and Clinical_Pick_Tier1.Level_4 have been downloaded from The Genome Data Analysis Centers (GDACs) (https://gdac.broadinstitute.org/) (Accessed date: 15 December 2020). Based on histological type, 95 « esophagus squamous cell carcinoma » have been selected and 11 « Normal Solid Tissue ». Total raw counts were then loaded on degust 4.1.1 [20]. All analyses were performed using EdgeR, TMM normalization and “Min gene read count” set at 10. Normal Solid Tissue samples were used as reference to calculate the fold change of gene expressions. To have an overview of clock gene expressions in different clinical variables, boxplot have been realized using CPM value of gene of interest and “pathologic stage” or “pathology N stage”. To clarify the tendency per “pathologic stage” and increased the number of sample per stage, we removed the letter “a”, ”b”, and ”c” and grouped the pathologic stage in “i”, “ii”, or “iii”. Finally, we applied Kruskal–Wallis test to highlight significant differences between the clinical groups. When a significant value was return, the pairwise multiple tests Tukey–Kramer have been performed to identify which group was different from the other. All processing of the data and analyses have been done with R version 3.6.3.

### 2.2. Cell Culture

Human eSCC lines KYSE-140 (ACC348), KYSE-180 (ACC379) and KYSE-410 (ACC381) were cultured in RPMI 1640 medium (Gibco/Thermo Fisher Scientific, Waltham, MA, USA, 21875-034) supplemented with 10% fetal bovine serum (Gibco, 10270-106) and 1% penicillin/streptomycin (Gibco, 15070-063). Human eSCC lines KYSE-270 (ACC380) and KYSE-450 (ACC387) were cultured in 49% RPMI 1640 and 49% Ham’s F12 (Gibco, 11765-054) supplemented with 2% fetal bovine serum and 1% penicillin/streptomycin. All KYSE lines were obtained from DSMZ (German Collection of Microorganisms and Cell Cultures Gmbh, Braunschweig, Germany) and cultured as suggested by the company. Primary human esophagus epithelial cells (H-6046) were obtained from CellBiologics (Chicago, IL, USA) and cultured in complete epithelial cell medium (H6621) as suggested by the company. Primary culture of FACS sorted YFP+ murine eSCC cells, were cultured in RPMI 1640 supplemented with 10% fetal bovine serum and 1% penicillin/streptomycin. All cells were cultured under humidified atmosphere with 5% CO_2_.

### 2.3. 4-NQO Mediated Induction of eSCC in Transgenic Mice

4-NQO (Sigma-Aldrich/Merck KGaA, Darmstadt, Germany, N8141) was dissolved in propylene glycol to a final concentration of 5 mg/mL and stored at −20 °C until use. When needed, this stock solution was further diluted 1/50 in distilled, sterile water to a final 4-NQO concentration of 100 mg/L.

The expression of the genetic construction K5CreER:p53fl/fl:RosaYFP was activated through intraperitoneal injection of 2.5 mg of tamoxifen (BOC Science, Shirley, NY, USA, B0084-358326) in sterile sunflower seed oil in 8 weeks-old mice housed under SOPF conditions. One week after injection, the drinking water of the mice was replaced by the aqueous 4-NQO solution, in opaque bottles, for 8 weeks (changing the solution twice a week). After the treatment, distilled, sterile drinking water was restored, and the animals were weighed weekly and carefully inspected daily.

### 2.4. Tissue Digestion and FACS Isolation

4-NQO-induced esophagus SCC were dissected, minced and digested in 2 mg/mL of collagenase I (A. G. Scientific, San Diego, CA, USA, C-2823) for 1 h. Collagenase I activity was blocked by the addition of 5 mM EDTA and incubation for 15 min. Trypsin (0.125%) (Capricorn Scientific, Ebsdorfergrund, Germany, TRY-2B10) was then added for 15 min and then the cells were rinsed in PBS supplemented with 2% FBS. All incubations were done on a rocking plate at 37 °C.

Immunostaining was performed on single cell suspension using PE-conjugated anti-CD45 (1:500, BioLegend, San Diego, CA, USA), PE-conjugated anti CD31 (1:500, BioLegend), PE-conjugated anti-CD140a (1:500, BioLegend) and Hoechst 33258 (1:10,000, Molecular Probes/Thermo Fisher Scientific, Waltham, MA, USA, H3569), during 45 min at 4 °C on a rocking plate. Living epithelial cells were selected by forward scatter, side scatter, doublets discrimination and by Hoechst dye exclusion. YFP+/Lin- cells were selected based on the expression of YFP and the exclusion of CD45, CD31, CD140a (Lin-). Control esophageal cells were selected based on the expression of APC-Cy7 EpCam (1:250, BioLegend) and the exclusion of CD45, CD31, CD140a (Lin-). Fluorescence-activated cell sorting analysis was performed using FACSAria III and FACSDiva software (BD Biosciences, San Jose, CA, USA).

### 2.5. Circadian Cycle Synchronization

Human KYSE-410 and murine eSCC cell lines were synchronized in the circadian cycle by addition of 1 µM dexamethasone (Sigma-Aldrich, D1756) for 1 h. Cells were then collected or treated 24 h post synchronization, at the top of PER2 expression, and 36 h post-treatment, at the lowest PER2 expression level. Mouse esophageal SCC cells were subjected to an additional cycle being also collected or treated 48 (high PER2 expression) and 60 (low PER2 expression) hours post-synchronization.

### 2.6. Bioluminescence Measurement and Data Analysis

Esophageal cancer cells (KYSE-410 and 4-NQO) harbored a rapidly-degradable form of luciferase, dLuc, driven by the PER2 gene promoter through lentiviral infection (PER2-dLuc) [21] purchased from DNA/RNA Delivery Core, Northwestern University, using Polybrene Infection Reagent (Millipore, Burlington, MA, USA TR1003G). The cells were maintained in a culture medium containing RPMI-1640 supplemented with 10% heat-inactivated FBS, 2 mM L-Glutamine, antibiotic antimycotic (all products from Life Technologies—Thermo Fisher Scientific, Waltham, MA, USA) and 2 μg/mL blasticidin (Sigma-Aldrich, 15205) to select for stable PER2-dLuc integration. Selected cells were then trypsinized (Trypsin-EDTA, Capricorn Scientific, TRY-1B10) and seeded in 3.5 cm dishes (1 × 10^5^ cells/cm^2^ for luciferase assay the following day). Cells were synchronized with 1 μM dexamethasone for 1 h. Cells were washed twice with PBS and the medium was replaced with 1.2 mL of phenol-red free medium supplemented with 352.5 μg/mL of sodium bicarbonate, 10 mM HEPES, 2 mM L-Glutamine, 5% FBS, 25 units/mL penicillin, and 25 μg/mL streptomycin (all products from Life Technologies), and 0.1 mM luciferin potassium salt (Biosynth AG, Staad, Switzerland, L-8240). Sealed cultures (VWR, cover glass 631-0177; Sigma-Aldrich Dow corning high-vacuum silicone grease, Z273554-EA) were placed in a Kronos Dio luminometer (Atto, Tokyo, Japan, AB-2550) at 37 °C. Bioluminescence was recorded continuously for 5 days; data were analyzed with the Kronos Dio software. Data were acquired and graphed using a custom-made script in RStudio. To measure the PER2-dLuc luminescence period from the time-series datasets, the “meta2d” function from R’ “MetaCycle” package (https://CRAN.R-project.org/package=MetaCycle) (Accessed date: 1 February 2019) [22] was used.

### 2.7. Cisplatin Treatment

Cisplatin EC_50_ value for the human KYSE-410 cell line was determined by incubating the cells in complete RPMI 1640 with 2.5 to 50 µM of cisplatin (Selleckchem, Munich, Germany, S1166) for 48 h at 37 °C. Cell viability was measured by MTS assay (Promega, Madison, WI, USA, G3581) 48 h after treatment following the instructions from the manufacturer.

Both cell lines were treated by the addition of 50 µM cisplatin to complete RPMI 1640 medium. Cells were incubated in the presence of cisplatin or the vehicle (NaCl) for 4 h at 37 °C.

### 2.8. Annexin V FACS Analysis

48 h after chemotherapy, the culture medium was removed and the cells were trypsinized (trypsin-EDTA). The removed medium was used to stop the trypsinization ensuring that the whole cell population was recovered. Cells were pelleted by centrifugation at 600× *g* for 5 min and the supernatant was discarded. The pellets were resuspended in 1X HBSS buffer (Gibco, 14025-050) containing 1:100 of Annexin V APC (Invitrogen—Thermo Fisher Scientific, Waltham, MA, USA, 17-8007-74) and 1:10,000 of Hoechst 33258 dye. After incubation at room temperature on a rocking platform for 30 min in the dark, the cells were further diluted in 1x HBSS and analyzed. FACS analysis was performed using a BD LSRFortessa X-20 and FACSDiva software (BD Biosciences).

### 2.9. Immunofluorescence Microscopy

eSCC cells were directly fixed with 10% neutral buffered formalin (VWR, 11699455) for 5 min at room temperature. After 3 washes of 5 min with PBS, all fixed cells were blocked with PBST-BSA [PBS containing 0.2% Triton X-100, 2% BSA (Sigma-Aldrich, A7906)] supplemented with 10% normal horse serum (Capricorn Scientific, HOS-1B). Primary antibodies (GFP from chicken, GenScript, A01694-40, 1:500; Krt 14 from chicken, Biolegend, 906004, 1:4000; p63 from rabbit, Abcam, ab124762, 1:1000) were diluted in PBST-BSA and incubated overnight at 4 °C. Cells were washed 3 times with PBS for 5 min and incubated with secondary antibody (Jackson Immunoresearch, West Grove, PA, USA, 1:500) and 1:1,000 of Hoechst dye for 1 h in the dark at room temperature. After washing, samples were mounted with Immu-Mount (Thermo Fisher Scientific, 9990402). Images were acquired using an Epifluorescence Zeiss Axioimager Z1 microscope equipped with Zeiss AxioCam MRc5 and a 20x objective (Plan-Neofluar, 20×/0.5; Zeiss, Jena, Germany). Acquired CZI images were then processed using ImageJ.

### 2.10. Immunoblot

Cells were lysed in Laemmli 1x sample buffer. Lysates were centrifuged for 10 min at 13,000 RPM at RT. 10-30 μg of protein extract were separated on polyacrylamide gels and transferred onto polyvinylidene difluoride membranes (PVDF, Millipore, Burlington, MA, USA, IPVH0010) using a wet system. The membranes were blocked in tris-buffered saline (TBS) with 0.1% Tween20 and 5% skim milk (VWR, 84615). Blots were probed with the appropriate primary antibodies overnight at 4 °C, followed by secondary goat anti-mouse (Cell Signaling Technologies, 7076) or anti-rabbit antibodies (Cell Signaling Technologies, 7074) conjugated to horseradish peroxidase and developed using enhanced chemiluminescence (PerkinElmer, Waltham, MA, USA, NEL104001EA). Images were obtained on a Vilber Lourmat Fusion Solo S. Primary antibodies used: Anti-PER2 (Abcam, Cambridge, United Kingdom, ab180655) and anti-γH2AX (p Ser139) (Cell Signaling Technologies, Danvers, MA, USA, 9718).

### 2.11. mRNA Extraction and RT-qPCR

RNA extraction was performed using the MicroElute Total RNA kit (Omega Biotek, Norcross, GA, USA, R6831-02) according to the manufacturer’s recommendations. Purified RNA was used to synthesize the first strand cDNA using Superscript II (Invitrogen) and random hexamers. RT-qPCR analyses were performed with 4 ng of cDNA as template, using a SYBR Green Mix (Applied Biosystems, Foster City, CA, USA, 100029284) and a QuantStudio 3 (Applied Biosystems) real-time PCR System.

Primers:Ms_Actb Fw(5->3) CACTGTCGAGTCGCGTCC Rv(5->3) TCATCCATGGCGAACTGGTGMs_Per2 Fw(5->3) CCACTATGTGACAGCGGAGG Rv(5->3) CTCTGGAATAAGCGCTTCGC Ms_Arntl Fw(5->3) GAGCGGATTGGTCGGAAAGTA Rv(5->3) TCTTCCAAAATCCAATGAAGGC

Relative quantitative RNA was normalized using the housekeeping gene β-actin. Primers were designed using NCBI Primer-BLAST (http://www.ncbi.nlm.nih.gov/tools/primer-blast/) (Accessed date: 1 December 2020). Analysis of the results was performed using QuantStudio Design and Analysis Software v1.4 (Applied Biosystems) and relative quantification was performed using the DDCt method using β-actin as reference.

### 2.12. RNAseq and Analysis of Bulk Samples

RNA quality was checked using a Fragment Analyzer 5200 (Agilent technologies, Santa Clara, CA, USA). Indexed cDNA libraries were obtained using the Ovation Solo RNA-Seq Library Preparation Kit (Tecan, Männedorf, Switzerland) following manufacturer’s recommendations. The multiplexed libraries were loaded on a NovaSeq 6000 (Illumina, San Diego, CA, USA) using a S2 flow cell and sequences were produced using a 200 Cycle Kit. Paired-end reads were mapped against the mouse reference genome GRCm38 using STAR software (version 2.5.3a) to generate read alignments for each sample. Annotations Mus_musculus.GRCm38.90.gtf were obtained from ftp.Ensembl.org (accessed on 15 December 2020). For human KYSE lines, paired-end reads were mapped against the human reference genome GRCh38 using STAR software (version 2.5.3a) to generate read alignments for each sample. Annotations Homo_sapiens.GRCh38.90.gtf were obtained from ftp.Ensembl.org (accessed on 15 December 2020). After transcripts assembling, gene level counts were obtained using HTSeq-0.9.1 [23]. Total raw counts were loaded on degust 4.1.1 [20]. All analyses were performed using EdgeR, TMM normalization and “Min gene read count” set at 10. Control esophagus samples were used as reference to calculate the fold change of gene expressions. Volcano plots were generated using the package “EnhancedVolcano” [24] from Bioconductor in R version 3.6.3.

### 2.13. Gene Set Enrichment Analysis (GSEA) Analysis

GSEA analysis was performed using preranked gene set enrichment analysis from the fgsea package [25] in R version 3.6.3, with “nperm = 1000” and “maxSize = 500”. The values of LFC were used as the ranking metric. For this purpose, thresholds were enlarged to include more genes in the analysis and therefore increase the strength of the analysis. The human C5 collection which contains gene sets annotated by GO terms have been downloaded (http://bioinf.wehi.edu.au/software/MSigDB/ [3]) (Accessed date: 10 January 2021) and all gene sets have been tested. For GSEA analysis, ranked fold change values correspond to 36 h over 24 h post-synchronization (abs(LFC) > 1, *p*-value < 0.05).

## 3. Results

### 3.1. Expression of Clock-Related Genes Is Altered in Human eSCC Samples

First, to determine putative alteration of the clock related genes in eSCC, we have analyzed publicly available data from the Cancer Genome Atlas (TCGA). Comparison between the 95 available eSCC samples and the 11 samples of normal esophagus unveiled a significant downregulation of CLOCK as well as members of the Period (PER) and Cryptochrome (CRY) families (Figure 1A). Other transcripts were either not modified (*ARNTL*, *NR1D2* and *RORA*) or upregulated (*NR1D1* and *TIMELESS*). Then, we aimed at characterizing the impact of cancer progression on the pattern of clock related genes. To this end, we sorted TCGA eSCC samples based on their pathological classification (i, ii, iii, and iv). To summarize, stage i describes small tumors without local invasion (*n* = 7), stage ii (*n* = 55) and iii (*n* = 27) describe invasive cancers potentially invading lymph nodes, and stage iv describes cancers with distant metastases (*n* = 4). We have investigated the expression of *PER1, PER2, PER3, CRY1, CRY2, ARNTL* (BMAL1), *TIMELESS, NR1D1* (REVERBα), *NR1D2* (REVERBβ), *CLOCK,* and *RORA* (RORα) in samples of eSCC at different stages (Figure 1B). We did not observe a significant downregulation of clock related genes in advances stages as compared to small, localized tumors.

A detailed representation of the expression of these genes depending on pathological classification is provided in the Appendix A. In the same line, we did not find a significant downregulation of clock related genes depending on the criterion of lymph node invasion (pathological N stage) (Appendix A).

### 3.2. PER2 Expression Oscillates in Human Esophageal SCC

To determine the pattern of clock related genes more specifically in esophageal SCC epithelial cells, we have used RNA sequencing to profile a commercially available non-immortalized human esophagus epithelial cell line as well as 5 different esophageal SCC cell lines from the KYSE series: KYSE-140, −180, −270, −10 and −450 [20] (Figure 2A). Using a threshold of false discovery rate (FDR < 0.01), we could find several deregulated genes in cancer cells (Figure 2B–F). Interestingly, instead of a global downregulation of all clock related genes as previously reported in human SCC samples from different organs [19], we found both upregulated and downregulated transcripts in esophageal cancer cell lines. This pattern also differed from the one found in the eSCC samples from the TCGA. Moreover, the 5 eSCC lines are different from each other. For instance, we observed a significant upregulation of PER2 and PER3 in 4 lines out of 5. However, other mRNAs such as *ARNTL* (coding for BMAL1 protein) is upregulated in some lines (KYSE-270 and −450), downregulated in another (KYSE-410 and 180) or not modified (KYSE-140) (Figure 2B–F).

To determine whether esophageal cancer cells have somehow conserved a circadian rhythm in spite of these modifications, we then focused our analysis on the KYSE-410 eSCC line. KYSE-410 cells are established from the poorly differentiated invasive esophageal squamous cell carcinoma resected from the cervical esophagus of a 51-year-old Japanese man prior treatment [26]. Co-immunostaining showed that these cells are characterized by p63 and Krt14 expression as expected from eSCC cells (Figure 2G). We transduced a PER2: Luc construct in KYSE-410 cells using lentiviral particles (Figure 2H). In this model, a 1-h dexamethasone treatment is sufficient to synchronize the cells and trigger oscillations of the luciferase activity that can be monitored for several days (Figure 2I). These results are consistent with those recently published with another cell line from the KYSE series [19]. Our results thus suggest that in spite of an altered expression of the clock related genes, the circadian oscillation of PER2 expression is still active in human eSCC cells.

### 3.3. The Level of PER2 Expression Is Associated to Transcription Modifications in Esophageal Cancer Cells

To determine the impact of PER2 oscillations on esophageal cancer cells, we have synchronized the KYSE-410 cells using dexamethasone and profiled the cells when PER2 expression is high or low (24 and 36 h after synchronization respectively) (Figure 3A). We found hundreds of genes significantly deregulated between these 2 time points (Log2 Fold of change > 0.65 and *p*-val <0.05), 482 transcripts were upregulated at 36 h as compared to 24 h, and 442 were downregulated (Figure 3B). Of note, PER1 was the most significant downregulated transcript at this timepoint. Gene set enrichment analysis (GSEA) suggested that multiples processes were impacted depending on PER2 expression (Figure 3C,D). Notably, we found that “*regulation of nucleotide metabolic process*” and “*regulation of cyclic nucleotide metabolic process*” were among the most significant negatively enriched gene sets, thus indicating that transcripts related to these biological processes were downregulated when PER2 expression is low. Inversely, “*apoptotic signaling pathway*” gene set related transcripts were enriched when PER2 expression is low (Figure 3C,D). These data suggest that depending on the circadian clock and PER2 expression, eSCC cells may be more or less sensitive to pro-apoptotic signals. These results are in line with a previous work that showed that CRY1/2 and PER1 influence the extrinsic TNFα-dependent pathway and intrinsic apoptotic pathways in cancer cells [15].

### 3.4. Efficiency of Cisplatin-Induced Apoptosis Depends on the Level of PER2 Expression

To determine whether the oscillations of PER2 expression might indeed regulate apoptosis in eSCC cells, we have treated cells with cisplatin. First, we have tested several concentrations of cisplatin on KYSE-410 cells to assess their sensitivity to the drug. We have treated KYSE-410 cells for 48 h with different concentrations of cisplatin (from 2.5 to 50 μM) and measured cell survival using MTS assay (Figure 4A). This experiment indicated that the EC_50_ for cisplatin in KYSE-410 cells is around 20 μM (Figure 4B). Since we had to treat eSCC cells for a short period of time (4 h), we have chosen to use a concentration that was sufficient to kill the majority of KYSE-410 cells (50 μM). We treated KYSE-410 cells with cisplatin for 4 h at either 24 (PER2 high expression) or 36 h (PER2 low expression) after dexamethasone synchronization (Figure 4C). We validated by immunoblotting that PER2 was indeed downregulated at 36 h compared to 24 h after synchronization (Figure 4D).

To determine whether a difference in the level of PER2 expression would have consequences on the global response to therapy, we then investigated whether cisplatin-induced DNA damage would be modified depending on the level of PER2 expression. To this end, we harvested KYSE-410 cells after a 4-h cisplatin treatment at either 24 or 36 h post-synchronization and measured γH2AX expression by Western blot (Figure 4E,F). This experiment shows that cisplatin induces more DNA damage when PER2 expression is low. We then measured apoptosis using annexin V staining by flow cytometry 48 h after the end of the treatment (Figure 4G). Interestingly, the percentage of eSCC cells positive for annexin V appeared significantly higher when cells were treated 36 h after synchronization, at the time when PER2 expression is low (Figure 4H). Hence, these results suggest that the circadian clock influences chemotherapy-induced cell death in esophageal cancer cells.

### 3.5. Characterization of the Pattern of Expression of Clock Related Genes in Mouse Esophageal SCC Primary Culture

In order to determine whether the pattern of clock related genes in biopsies of human eSCC would be affected by their contamination with non-epithelial cells, we have investigated clock genes in a mouse model of eSCC. To this end, we used a model of chemically-induced esophageal cancer based on the addition of 4-NQO (4-Nitroquinoline 1-oxide) in the drinking water. We treated K5CreER:p53fl/fl:RosaYFP (K5:p53:YFP) mice with tamoxifen to remove p53 and turn on the YFP fluorescent reporter in esophageal cells (Figure 5A). We treated 8 weeks-old K5:p53:YFP mice with 4-NQO over 8 weeks and then followed them over time until they developed tumors (Figure 5B). We could dissect individual tumors from K5:p53:YFP esophagus and digested these to isolate tumor epithelial cells by flow cytometry. We verified the histology of the primary tumors prior to sequencing. These tumors were characterized by classical eSCC markers p63 and Krt14 (Figure 5C). FACS analysis confirmed the expression of YFP by the tumor cells, allowing us to sort pure YFP+ tumor epithelial cells and get rid of the contamination with lineage negative cells (CD45+ leukocytes, CD31+ endothelial cells and CD140a+ fibroblasts) (Figure 5D). We profiled YFP+ eSCC epithelial cells by RNAseq and compared them to FACS sorted EpCam+ control esophageal epithelial cells (Figure 5D). We found 3,492 transcripts significantly deregulated in cancer cells (Log2 Fold of change >1 and FDR < 0.01). We focused our analysis on clock related genes and found no difference except for PER3, which was upregulated (LFC = 1.21) in cancer cells compared to normal cells and *Rora*, which appeared downregulated (LFC= −2.07) in cancer cells compared to control esophageal cells (Figure 5E).

### 3.6. A Low PER2 Expression Is Associated to a Higher Sensitivity to Cisplatin-Mediated Apoptosis in Mouse Esophageal SCC Cells

We grew FACS sorted YFP+ tumor epithelial cells in vitro (Figure 6A) and transduced the PER2:Luc construct in these cells using lentiviral particles. Similar to human eSCC cells, a 1-h dexamethasone treatment was sufficient to synchronize the cells and trigger luciferase daily oscillations that can be monitored on real time for several days (Figure 6B). These results therefore demonstrate that PER2 expression oscillates in primary cultures of mouse esophageal cancer epithelial cells. We then measured PER2 and *Arntl* expression by qPCR in mouse eSCC epithelial cells after synchronization (Figure 6C). This experiment confirmed the variations of PER2 expression measured using bioluminescence assay, although there is a 4-h delay between bioluminescence and mRNA expression. Nonetheless, we did not see a typical antiphase expression of *Arntl* in mouse eSCC cells, suggesting that the biological circadian clock would be altered in these cells and that only PER2 expression may still oscillate in this model.

To determine whether PER2 oscillations had an impact on response to therapy as we found out in human eSCC cells, we treated mouse tumor epithelial cells with cisplatin for 4 h when PER2 expression is high (24 and 48 h) or when it is low (36 and 60 h) (Figure 6D). We then measured cell death by counting the proportion of annexin V positive cells 48 h after the end of the treatment (Figure 6E). As we observed in human eSCC cells, the level of cisplatin-induced apoptosis was significantly higher when PER2 expression is low (at 36 and 60 h after synchronization) (Figure 6F). Hence, it seems that in both mouse and human eSCC cells, PER2 oscillations lead to variable sensitivity to chemotherapy (Figure 6G).

## 4. Discussion

Many studies have investigated the role of the circadian clock in cancer cells over the last 2 decades [16]. It is now clear that the disruption of the circadian rhythm increases the risk of developing cancer [8,9]. CLOCK and BMAL1 clock master genes may harbor tumor-suppressive functions both in humans and rodents [27]. However, depending on the model and the tissue, CLOCK and BMAL1 may be oncogenic or tumor suppressive [28,29,30,31]. Mouse models with an invalidation of the clock master genes PER1 and/or PER2 display an increased risk of spontaneous and radiation-induced cancer development [32,33,34]. In spite of the downregulation of clock related genes in cancer, the persistence of a circadian clock or at least of PER2 oscillations have been reported in several cancer models. Indeed, the use of PER2:Luc had been previously used to demonstrate this rhythmic expression of PER2 in low malignancy breast cancer cells [35], and more recently in cervical and esophageal cancer cell lines [19]. Our results unveil that human eSCC cell lines display various patterns of expression of clock related genes. This deregulation is not associated to a downregulation of the PER family members as observed in biopsies from human eSCC. In addition, our results also clearly evidence an oscillation of PER2 expression in esophageal squamous cell carcinoma (eSCC) cells from two different species. The difference between the pattern found in biopsies and tumor epithelial cells could be the consequence of the presence of non-tumoral cells in biopsies such as immune cells or cancer associated fibroblasts, but the status of clock-related genes in stromal cells should be thoroughly investigated to test this hypothesis. This difference might also reflect various stages of cancer progression. However, our data show that the tumor pathological stage is not associated to a significant modification of the clock-related genes expression. Since the number of eSCC samples is quite limited in the TCGA database, especially for metastatic cancers, further experiments will be required to determine how cancer progression impacts the expression of clock related genes.

Clock related genes are associated to multiple functions in cancer cells, such as cell-cycle, cell signaling, cell death, DNA damage response, or metabolism [16]. It has already been reported that clock disruption results in the accumulation of DNA damage. CRY2 expression is stimulated by DNA-damage and can regulate an intra-S checkpoint function in a ATR/CHK1 dependent manner [36]. PER1 interacts with ATM/CHK2 following radiation-induced DNA damage [37]. In the same line, PER2 downregulation would promote the resistance to radiation-induced apoptosis by delaying CHK2 expression [38]. In our model, we could show by RNA sequencing analysis that a low PER2 expression is associated to a downregulation of transcripts involved in “metabolism of nucleic acids”, and to an increase in “apoptotic signaling pathway” related genes. This pattern suggests that response to therapy may be different depending on the circadian rhythm, mostly because cells might accumulate more DNA damage and undergo apoptosis more frequently when PER2 expression is low. Consistently, mice homozygous for the mPer2^m/m^ mutation show an increased sensitivity to radiation and a higher risk to develop lymphoma [32], while PER1 is involved in the regulation of proapoptotic signals [15].

Cisplatin is one of the most common chemotherapeutic drugs. It is widely used to treat many solid tumors, including lung cancer, head and neck cancer and esophageal cancer [39]. Despite its effectiveness, many patients have intrinsic resistance to cisplatin or develop resistance to cisplatin over time. Platinum resistance thus remains a major issue in the therapy for many types of cancer. The mechanisms of resistance are complex and have not been fully elucidated. Among these mechanisms, enhanced DNA repair is considered as the most important cause of cisplatin resistance [39]. Indeed, cisplatin causes DNA intra- and interstrand cross-links by forming cisplatin-DNA adducts. These cisplatin-DNA adducts lead to damages that are repaired by nucleotide excision repair [40]. Interestingly, it has been reported that cisplatin-induced DNA damage is repaired by two circadian programs [41]. Our results show that cisplatin-induced DNA damage is enhanced when PER2 expression is low in human eSCC cells. It will be important to investigate the molecular mechanisms that link PER2 expression, DNA damage and apoptosis in esophageal cancer cells. Since we found that PER1 was the most significant downregulated genes when PER2 expression is low, the process may be related ATM/CHK2 as reported in a human colon cancer cell line [37]. Since the level of PER2 expression correlates with apoptosis related genes, it will be important to determine whether PER2 oscillatory expression sensitizes esophageal cancer cells to other anti-cancer therapies and whether PER1 is involved in this process.

The goal of chronotherapy is to deliver the drug at a time of the day that is optimal for killing cancer cells while avoiding resistance and/or toxicity [42,43]. Since cisplatin half-life in the blood stream is short (4h), administering this chemotherapy when the cells are the most sensitive to it can be relevant for its efficiency. In melanoma, the efficiency of cisplatin in PER1/2^−/−^ animals is higher than in control specimens [44]. In the same line, our results clearly indicate that cisplatin treatment of esophageal cancer cells is more efficient when PER2 expression is low. Previous studies already suggested that circadian rhythm may affect the response to therapy in esophageal cancer cells. Indeed, apoptotic regulators DEC1 and DEC2 repress CLOCK/BMAL-induced transactivation of the PER1 promoter and modulate the response to cisplatin in several types of cancers [45,46,47], including human eSCC cell lines [48,49]. Nonetheless, a treatment based on physiological oscillations of PER2 has never been tested before in eSCC. Our results therefore suggest that chronotherapy could be used to improve the efficiency of current chemotherapy in esophageal cancers. Several attempts have been made to develop chronotherapy regimens for combinations of drugs that included platinum salts [50,51]. Nevertheless, the gains have been either minor or have not been replicated in further studies [52]. Hence, cisplatin chronotherapy is not practiced in clinical oncology. Further characterization of the process by which circadian clock modulates chromatin stability and genome maintenance will be required. Indeed, although there are growing evidences of connections between circadian rhythm and genome maintenance [53], the underlying mechanisms are still largely unknown.

To determine whether chronotherapy might be used to improve treatment of esophageal cancers, the characterization of PER2 oscillations in eSCC cells in vivo will be required. Then the efficiency of cisplatin treatment based on the time of the day and/or PER2 expression should be tested in an in vivo setting such as xenografts of multiple eSCC lines and/or patient derived xenograft (PDX). Since cancer progression is associated to reduced BMAL1 and PER2 oscillations, it will be important to determine whether more malignant esophageal cancer epithelial cells are characterized by additional modifications of clock related genes and how the circadian clock is impacted. In conclusion, our results reveal that in spite of alterations of the expression of clock related genes, PER2 oscillations could be used to improve response to chemotherapy in esophageal cancer.

## Figures and Tables

**Figure 1 biology-10-00266-f001:**
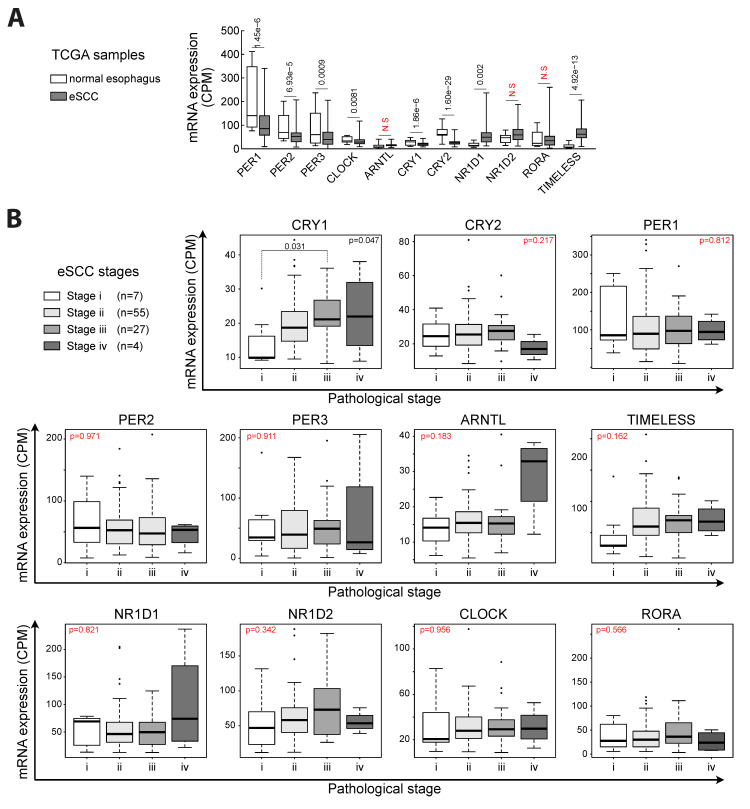
*Expression of clock-related genes in human esophageal squamous cell carcinoma (eSCC) samples.* (**A**) Expression of clock related genes measured by RNA sequencing in biopsies from control esophagus (*n* = 11) and eSCC (*n* = 95). Data are represented as boxplots. False discovery rate (FDR) is indicated for each gene. Benjamini–Hochberg method on the *p*-values was used to control the FDR. (**B**) Expression of clock related genes measured by RNA sequencing in biopsies from eSCC samples depending on their pathological stage. Data are represented as boxplots. Statistics were calculated using a Kruskal–Wallis test followed by a Tukey–Kramer test. *p*-values < 0.05 are considered as significant.

**Figure 2 biology-10-00266-f002:**
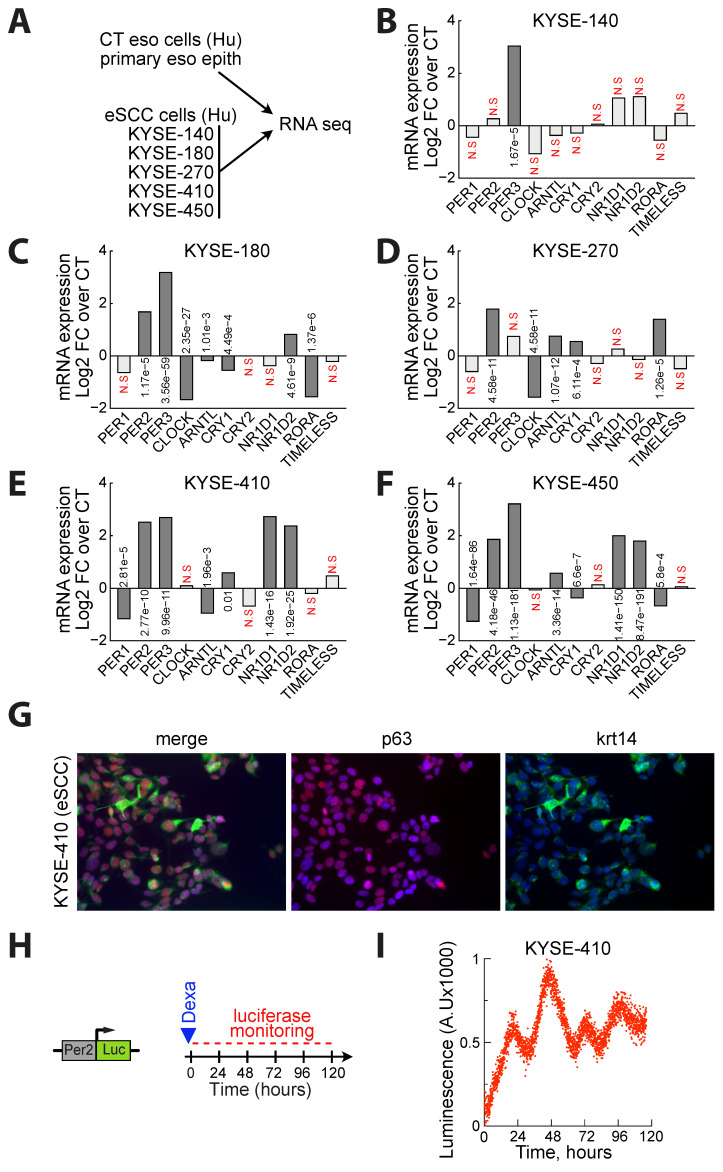
*PER2 expression oscillates in human eSCC cells in spite of alterations in the pattern of clock-related gene expression.* (**A**) Experimental design. (**B**) Histogram depicting the expression of clock related genes measured by RNA sequencing in the KYSE-140 human eSCC cell line. Data are represented as the mean of a biological duplicate. FDR is indicated for each gene. Benjamini–Hochberg method on the *p*-values was used to control the false discovery rate (FDR). (**C**) Same as in (B) in the KYSE-180 human eSCC cell line. (**D**) Same as in (**B**) in the KYSE-270 human eSCC cell line. (**E**) Same as in (B) in the KYSE-410 human eSCC cell line. (**F**) Same as in (B) in the KYSE-450 human eSCC cell line. (**G**) Co-immunostaining of p63 and Krt14 in the KYSE-410 human eSCC cell line. (**H**) Experimental design to monitor the activity of the PER2 promotor in KYSE-410 cells. (**I**) Real time monitoring of luciferase activity in the KYSE-410 human eSCC cell line. FDR>5% are considered non-significant (N.S).

**Figure 3 biology-10-00266-f003:**
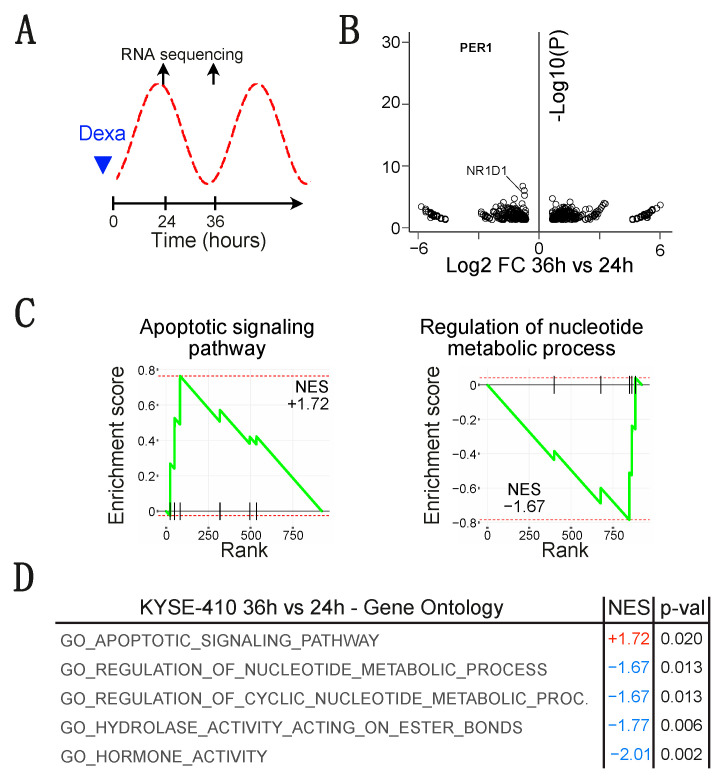
*PER2 expression oscillations are associated to modifications of the transcriptome.* (**A**) Experimental design. (**B**) Volcano plots representing results of RNA-seq as the statistical significance versus the magnitude of fold of change (FC) in KYSE-410 compared to control esophageal cells. Data are filtered based on FC (abs(LFC) > 1) and *p*-value (*p* < 0.05). (**C**) Gene set enrichment analysis (GSEA) of the significantly modified transcripts in KYSE-410 cells compared to control esophageal cells (abs(LFC) > 1, *p*-value < 0.05). (**D**) Table summarizing the results of the GSEA. Pathways with a *p*-value < 0.05 are listed and sorted based on normalized enrichment score.

**Figure 4 biology-10-00266-f004:**
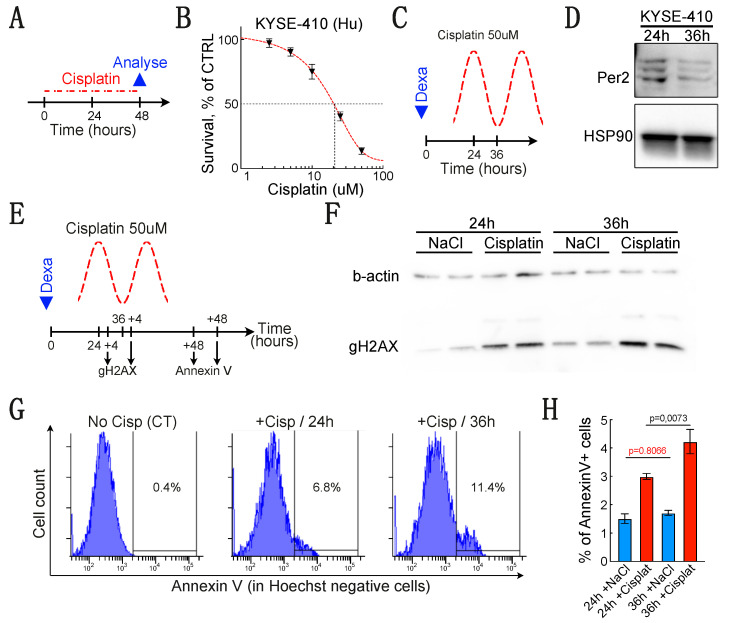
*A low PER2 expression is associated to higher sensitivity to cisplatin-induced apoptosis.* (**A**) Experimental design. (**B**) KYSE-410 survival measured using MTS assay 48 h after cisplatin treatment. The red sigmoid curve is a guide for the eye. Data are represented as mean +/− SEM (*n* = 5). (**C**) Experimental design to treat KYSE-410 when PER2 expression is high or low (24 or 36 h after synchronization respectively). (**D**) Western blot showing the expression of PER2 24 or 36 h after synchronization. (**E**) Schematic representation of the cisplatin treatment timeline and the time-points selected to collect the samples for DNA damage (γH2AX) evaluation and apoptosis (annexin V) measurement. (**F**) Western blot showing the expression of γH2AX after a 4-h cisplatin treatment either 24 or 36 h post-synchronization. Cells were either treated with vehicle alone (NaCl) or with 50 μM cisplatin. (**G**) Representative histogram showing the proportion of Hoechst-/Annexin V KYSE-410 cells measured by flow cytometry 48 h after the end of the cisplatin treatment. Cells were either not treated (NaCl = vehicle) or treated for 4 h with 50 μM cisplatin 24 or 36 h after synchronization. (**H**) Histogram summarizing the proportion of Hoechst-/Annexin V KYSE-410 cells measured 48h after the end of the cisplatin treatment. Data are represented as mean +/− SEM (*n* = 8). *p*-values were calculated using a two-way ANOVA followed by Sidak’s post hoc test. Fixed effect: Time/*p* = 0.0110; Cisplatin/*p* < 0.0001; Time x Cisplatin/*p* = 0.0527.

**Figure 5 biology-10-00266-f005:**
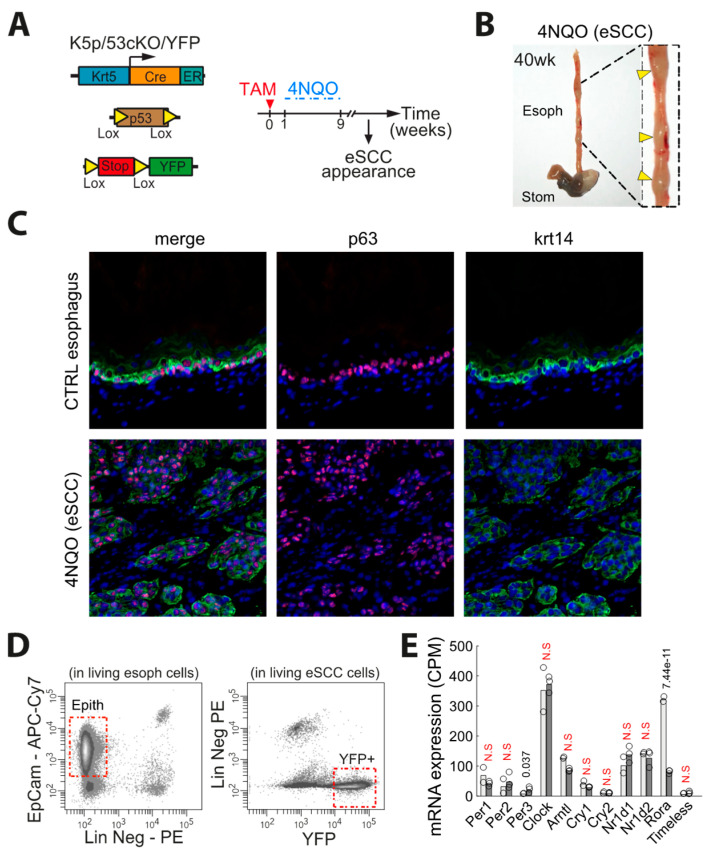
*PER2 expression oscillates in primary culture of mouse eSCC cells and influences sensitivity to chemotherapy.* (**A**) Genetic strategy and experimental design. (**B**) Representative picture of 4-NQO-induced eSCC in K5:p53:YFP mouse esophagus. (**C**) Co-immunostaining of p63 and Krt14 in normal esophagus and 4-NQO-induced eSCC. (**D**) FACS plot showing the expression of EpCam and lineage negative markers (CD45, CD31, and CD140a) in control esophageal cells, and YFP and lineage negative markers in 4-NQO-induced eSCC. (**E**) Expression of clock-related genes measured by RNA sequencing in FACS sorted Epcam+ normal esophagus epithelial cells (*n* = 2) and YFP+ 4-NQO-induced eSCC epithelial cells (*n* = 3). FDR is indicated for each gene. Benjamini–Hochberg method on the *p*-values was used to control the false discovery rate (FDR). FDR>5% are considered non-significant (N.S). “Epith” means epithelial cells.

**Figure 6 biology-10-00266-f006:**
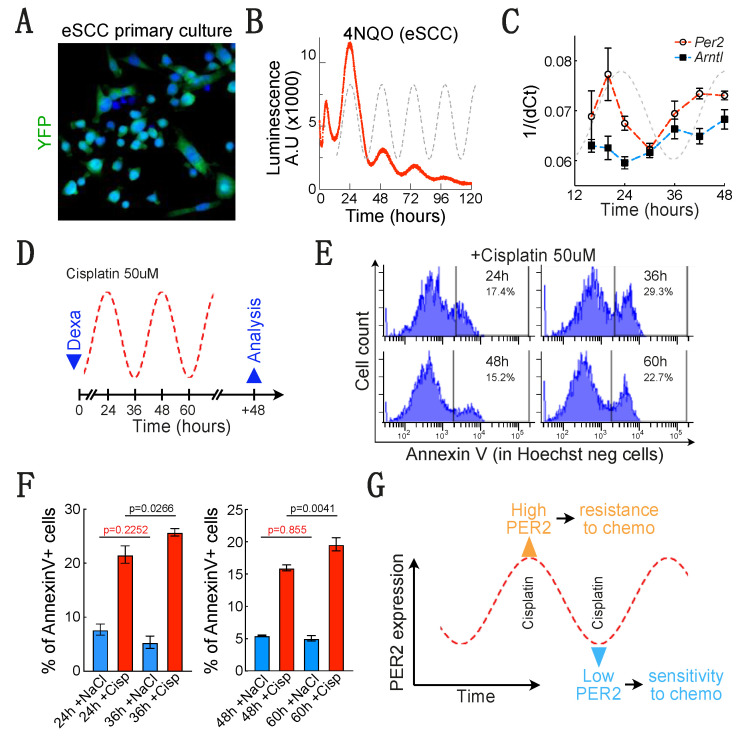
*A low PER2 expression is associated to a higher sensitivity to cisplatin in mouse eSCC epithelial cells.* (**A**) Endogenous YFP fluorescence in primary culture of 4-NQO-induced eSCC epithelial cells. (**B**) Real time monitoring of luciferase activity in primary culture of 4-NQO-induced eSCC epithelial cells. (**C**) PER2 and *Arntl* expression on the 4-NQO induced eSCC epithelial cells measured by qPCR for 48 h after dexamethasone-induced synchronization. (**D**) Experimental design to treat 4-NQO-induced eSCC epithelial cells when PER2 expression is high or low (24 or 36 h after dexamethasone-induced synchronization). (**E**) Representative histogram showing the proportion of Hoechst-/Annexin V 4-NQO-induced eSCC epithelial cells measured by flow cytometry 48 h after the end of the cisplatin treatment. Cells were either not treated (NaCl = vehicle) or treated for 4 h with 50 μM cisplatin 24, 36, 48 or 60 h after synchronization. (**F**) Histogram summarizing the proportion of Hoechst-/Annexin V+ 4-NQO-induced eSCC epithelial cells measured 48 h after the end of the cisplatin treatment. Data are represented as mean +/− SEM. *p*-values were calculated using a two-way ANOVA followed by Sidak’s post hoc test. First cycle (*n* = 8). Fixed effect: Time/*p* = 0.3788; Cisplatin/*p* < 0.0001; Time x Cisplatin/*p* = 0.0067. Second cycle (*n* = 5). Fixed effect: Time/*p* = 0.0231; Cisplatin/*p* < 0.0001; Time x Cisplatin/*p* = 0.0231. (**G**) Scheme depicting our main observations in esophageal cancer cells.

## Data Availability

The results shown here are in whole or part based upon data generated by the TCGA Research Network: https://www.cancer.gov/tcga (accessed date: 15 December 2020).

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
