# Peer review of "PER2 Circadian Oscillation Sensitizes Esophageal Cancer Cells to Chemotherapy"

_biology, 2021, doi:10.3390/biology10040266_

Round 1

Reviewer 1 Report

The authors Redondo et al in their paper titled ‘’A circadian oscillation of Per2 sensitizes esophageal cancer cells to chemotherapy” have suggested by their observations in cisplatin based chronotherapy for the treatment of esophageal cancer. The authors have systematically established the existence of a circadian oscillation in the tumors and also a regimen that best affects administration of Cisplatin or in extension chemotherapy. 

There is only one minor comment if the authors can show pictorially a summary of the overall structure of the study as to when the best time to treat cells would be. This is not required but in a glance would show the significance of the study.

As a curiosity have the authors looked into the cells as PDX models to see if chronotherapy regimens hold good in the PDX models

Author Response

Reviewer#1

The authors Redondo et al in their paper titled ‘’A circadian oscillation of Per2 sensitizes esophageal cancer cells to chemotherapy” have suggested by their observations in cisplatin based chronotherapy for the treatment of esophageal cancer. The authors have systematically established the existence of a circadian oscillation in the tumors and also a regimen that best affects administration of Cisplatin or in extension chemotherapy. 

There is only one minor comment if the authors can show pictorially a summary of the overall structure of the study as to when the best time to treat cells would be. This is not required but in a glance would show the significance of the study.

We would like to thank the Reviewer#1 for his/her fair assessment of our work and for his/her suggestions.

As suggested by the Reviewer#1, we have included a scheme summarizing our observations and pointing to the best expected time to treat esophageal cancer cells with cisplatin. This scheme appears in the revised version of our manuscript as the Figure6G.

Q1. As a curiosity have the authors looked into the cells as PDX models to see if chronotherapy regimens hold good in the PDX models?

A1. This is a very good point raised by the Reviewer#1. This is clearly a very important experiment to do in the near future, but we have not tried it yet. To highlight the importance of this experiment, we now mention PDX in the revised version of our discussion: “Then the efficiency of cisplatin treatment based on the time of the day and/or PER2 expression should be tested in an in vivo setting such as xenografts of eSCC lines and/or patient derived xenograft (PDX).” (Lines 581-583).

Reviewer 2 Report

1. There are no messages from this paper other than data showing that Per2 expression levels are associated with sensitivity to cisplatin (Fig. 4 & 6). It has long been known that Per2 expression oscillates independently from BMAL1/CLOCK. It is therefore unclear whether this difference in the sensitivity is controlled by the biological clock or independently by PER2. The author should delve into this difference in sensitivity.

2. The term Oscillation should be used correctly. For example, the result in Fig. 1 is not oscillation but simply expression. Therefore, whether Per2 maintains oscillation in cancer cells is unclear from the data in this paper. We therefore cannot support our argument.

3. The discussion part is meaningless. The discussion should be based on the data.

Author Response

Reviewer#2

  1. There are no messages from this paper other than data showing that Per2 expression levels are associated with sensitivity to cisplatin (Fig. 4 & 6). It has long been known that Per2 expression oscillates independently from BMAL1/CLOCK. It is therefore unclear whether this difference in the sensitivity is controlled by the biological clock or independently by PER2. The author should delve into this difference in sensitivity.

We would like to thank the Reviewer#2 for his/her comments on our work.

We must confess that we do not understand the point of the Reviewer#2. We are aware of what has been previously shown in the literature and we have tried to give an overview of this field in our introduction and our discussion. The point of our study is to investigate the mechanisms associated to PER2 in esophageal cancer cells. Our messages can be summarized as following:

  1. Clock related genes are deregulated in esophageal cancer biopsies as compared to normal tissue.
  2. Tumor progression is not correlated to a significant repression of the clock related genes.
  3. Clock related genes are deregulated in esophageal cancer epithelial cells (eSCC) as compared to normal esophageal cells.
  4. In spite of the deregulated expression of several clock related genes, PER2 expression keeps oscillating in human eSCC cells (KYSE-410).
  5. PER2 expression levels are associated with sensitivity to cisplatin in esophageal cancer cells.
  6. In mouse eSCC cells, the clock related genes expression is not profoundly affected. Per2 expression oscillates in these cells but Arntl expression does not.
  7. The level of Per2 expression influences response to cisplatin in esophageal cancer cells in mouse.

We have carefully checked the literature and have not found these data in other articles except for the Figure1A, which is similar to the data published by Van der Watt et al., a few months ago and that partially scooped us indeed. One can easily look for articles on this topic and see that very little is known about PER2 in esophagus cancer. We have listed below the 9 articles on “PER2 + esophagus” published on Pubmed (March 7th 2021):

1: Dakup PP, Porter KI, Gajula RP, Goel PN, Cheng Z, Gaddameedhi S. The circadian clock protects against ionizing radiation-induced cardiotoxicity. FASEB J. 2020 Feb;34(2):3347-3358. doi: 10.1096/fj.201901850RR. Epub 2020 Jan 10. PMID: 31919902.

2: Hashimoto A, Uemura R, Sawada A, Nadatani Y, Otani K, Hosomi S, Nagami Y, Tanaka F, Kamata N, Taira K, Yamagami H, Tanigawa T, Watanabe T, Fujiwara Y. Changes in Clock Genes Expression in Esophagus in Rat Reflux Esophagitis. Dig Dis Sci. 2019 Aug;64(8):2132-2139. doi: 10.1007/s10620-019-05546-1. Epub 2019 Feb 28. PMID: 30815822.

3: Yang SC, Chen CL, Yi CH, Liu TT, Shieh KR. Changes in Gene Expression Patterns of Circadian-Clock, Transient Receptor Potential Vanilloid-1 and Nerve Growth Factor in Inflamed Human Esophagus. Sci Rep. 2015 Sep 4;5:13602. doi: 10.1038/srep13602. PMID: 26337663; PMCID: PMC4559770.

4: Kentish SJ, Frisby CL, Kennaway DJ, Wittert GA, Page AJ. Circadian variation in gastric vagal afferent mechanosensitivity. J Neurosci. 2013 Dec 4;33(49):19238-42. doi: 10.1523/JNEUROSCI.3846-13.2013. PMID: 24305819; PMCID: PMC6618780.

5: Weigl Y, Ashkenazi IE, Peleg L. Rhythmic profiles of cell cycle and circadian clock gene transcripts in mice: a possible association between two periodic systems. J Exp Biol. 2013 Jun 15;216(Pt 12):2276-82. doi: 10.1242/jeb.081729. Epub 2013 Mar 26. PMID: 23531816.

6: Castillo C, Molyneux P, Carlson R, Harrington ME. Restricted wheel access following a light cycle inversion slows re-entrainment without internal desynchrony as measured in Per2Luc mice. Neuroscience. 2011 May 19;182:169-76. doi: 10.1016/j.neuroscience.2011.02.003. Epub 2011 Mar 12. PMID: 21392557.

7: Guenthner CJ, Bickar D, Harrington ME. Heme reversibly damps PERIOD2 rhythms in mouse suprachiasmatic nucleus explants. Neuroscience. 2009 Dec 1;164(2):832-41. doi: 10.1016/j.neuroscience.2009.08.022. Epub 2009 Aug 19. PMID: 19698763; PMCID: PMC2762007.

8: Davidson AJ, Castanon-Cervantes O, Leise TL, Molyneux PC, Harrington ME. Visualizing jet lag in the mouse suprachiasmatic nucleus and peripheral circadian timing system. Eur J Neurosci. 2009 Jan;29(1):171-80. doi: 10.1111/j.1460-9568.2008.06534.x. Epub 2008 Nov 21. PMID: 19032592.

9: Molyneux PC, Dahlgren MK, Harrington ME. Circadian entrainment aftereffects in suprachiasmatic nuclei and peripheral tissues in vitro. Brain Res. 2008 Sep 4;1228:127-34. doi: 10.1016/j.brainres.2008.05.091. Epub 2008 Jun 14. PMID: 18598681.

As suggested by the Reviewer#2, some of our findings have been indeed described in other cancer models. But to the best of our knowledge, this is the first time that these results are described in esophageal cancer cells. Hence, we think that saying that “there are no messages” in our paper is unfair. This comment suggests that observations made in some types of cancers would be necessarily conserved in every single cancer model, including esophageal cancers. To our opinion, this is an overclaim and it might actually be completely untrue. Hence, the novelty of our work is related to the characterization of the role of PER2 in eSCC cells from 2 different species. We looked at the expression of clock related genes depending on cancer progression and we also measured the expression of these genes on FACS sorted tumor epithelial cells. This has never been done before in eSCC.

To add novelty to this study, we have now included a figure showing that PER2 expression levels influence DNA damage in human eSCC cells (Figure4F). This difference may at least partially explain the different levels of apoptosis measured after cisplatin treatment at 24 and 36 hours after synchronization. In addition, we also show that the level of apoptosis correlates with Per2 expression by comparing 24 and 36 h as well as 48 and 60 hours after synchronization (Figure6E,F). We have also included data on Arntl mRNA expression by qPCR after synchronization (Figure6C). Interestingly, we did not observe fluctuations of the expression of Arntl. These results suggest that variations in sensitivity to cisplatin are mostly associated to Per2 expression and not necessarily to the biological clock. We have made this clear in the revised version of our manuscript. Again, although this has been reported in other models, this is the first time that it is described in esophageal cancer cells.

  1. The term Oscillation should be used correctly. For example, the result in Fig. 1 is not oscillation but simply expression. Therefore, whether Per2 maintains oscillation in cancer cells is unclear from the data in this paper. We therefore cannot support our argument.

We have tried to use “oscillation” in a sense that the majority of the people would understand. According to the Oxford’s dictionary an oscillation is “a regular movement between one position and another or between one amount and another”. Hence, we think that a broad audience would perfectly understand that the luciferase assays illustrated in the Figures 2I and 6B show an oscillatory expression of PER2 in human and mouse esophageal cancer cells. Consistent with this notion, in a study on breast cancer by Lellupitiyage Don et al., one can read: “Subsequent luminometry experiments revealed that circadian oscillations of both BMAL1:luc and PER2:luc persisted with a typical antiphase relationship”. In the same line, in a study on cervical and esophageal cancers by Van der Watt et al., one can read: “Real-time monitoring of luciferase activity in additional cancer cell lines revealed distinct circadian oscillation of PER2-driven bioluminescence in all the cervical cancer (Fig. 5Di) and oesophageal cancer (Fig 5Dii) cell lines tested”. For these reasons, we think that saying that we observe an oscillatory expression of PER2 in our models, is not an overclaim. Nonetheless, we do not think that PER2 maintains oscillations of BMAL1/CLOCK in our system (Figure6C). To address the Reviewer#2 concern, we have therefore clearly described an “oscillatory expression of Per2” (Line 306). We also wrote clearly that our findings should be tested in an in vivo context (Lines 581-583).

  1. The discussion part is meaningless. The discussion should be based on the data.

The Reviewer#2 claims that there are “no messages” in our study. It is therefore logical that our discussion seems to be meaningless. We understand that our results cannot be considered as a breakthrough in the field of circadian rhythm, and that they are not pointing to brand-new concepts. Still, this is the first time that these results are reported in esophageal cancer cells. We have tried to clearly discuss our results, to compare them to what is currently known and to suggest future experiments that may help answering open questions. Of note, the 2 other reviewers did not comment the “emptiness” of our discussion, suggesting that it may be of interest for some readers.

Reviewer 3 Report

  1. Wouldn’t the upregulation and deregulation of other various clock components taking place in the cancer cells lead to changes in the sensitivity to chemotherapy through a different mechanism? For the mRNA expression, the most invasive stage of cancer (iv) shows even more significant differences when compared to local-level ones (i) especially in Cry1, Cry2, Per3, ARNTL, TIMELESS, and NR1D1 than Per2. How can you say that the altered expression of clock-related genes does not affect the sustained Per2 oscillation as well as other clock related activities in the esophageal cancer cells?
  2. As a follow-up, a detailed explanation with a diagram of the clock mechanism may be helpful, since the relationship between the oscillatory Per2 expression and chemotherapy need to be more clearly demonstrated.
  3. There are many carcinogenic cell lines, and they may differ functionally, structurally and genetically. Are you suggesting that this relationship between the Per2 expression and chemotherapy is universal among all mammalian esophageal cancers?
  4. You have the data for 24h and 36h after dexamethasone application. What about 48h and 60h? There needs to be at least two cycles of the oscillation.

Author Response

Reviewer#3

  1. Wouldn’t the upregulation and deregulation of other various clock components taking place in the cancer cells lead to changes in the sensitivity to chemotherapy through a different mechanism? For the mRNA expression, the most invasive stage of cancer (iv) shows even more significant differences when compared to local-level ones (i) especially in Cry1, Cry2, Per3, ARNTL, TIMELESS, and NR1D1 than Per2. How can you say that the altered expression of clock-related genes does not affect the sustained Per2 oscillation as well as other clock related activities in the esophageal cancer cells?

We would like to thank the Reviewer#3 for his/her comments on our work. As suggested by the Reviewer#3, we have measured Arntlexpression by qPCR after synchronization (Figure 6C) and did not observe an oscillatory expression as we did for Per2. These results suggest that variations in sensitivity to cisplatin are associated to Per2 expression mostly and not necessarily to the biological clock. In human eSCC cells, the pattern of clock related genes is profoundly modified (Figure2), this pattern is not modified in mouse eSCC cells as compared to normal epithelium. Still, the level of Per2 expression sensitizes eSCC cells to cisplatin-induced apoptosis in both species That is why we think that changes in the sensitivity to chemotherapy mostly relies on the level of Per2 expression. Since our RNAseq data show that Per2 expression is associated to the transcription of many genes, notably Per1, we cannot discard that Per1 is involved in the process, but that should be further investigated (Lines 554-557).

For the mRNA expression, we show in the Figure1B that the expression of CRY2, PER1, PER2, PER3, ARNTL, TIMELESS, NR1D1, NR1D2, CLOCK and RORA is not significantly modified along tumor progression. Although we can see an increase in the level of ARNTL and NR1D1, this difference is not significant probably because the number of samples for stage iv eSCC is low (n=4). Since, invasive eSCC are not surgically removed, it is difficult to have access to more material to determine whether this trend is really significant. We say that altered expression of clock-related genes does not affect sustained PER2 expression based on the data from the Figure2E and Figure2I. We can clearly see that clock-related gene expression is significantly modified in several eSCC cell lines, notably the KYSE-410. In the latter, we could successfully monitor an oscillatory PER2 expression. These results are in line with those from Van der Watt et al., on KYSE-30 published few months ago (Mol. Cancer Res. 2020). We have corrected our manuscript to make sure we do not claim that the clock related activities are not altered.

  1. As a follow-up, a detailed explanation with a diagram of the clock mechanism may be helpful, since the relationship between the oscillatory Per2 expression and chemotherapy need to be more clearly demonstrated.

As suggested by the Reviewer#3, we have included a scheme summarizing our observations and pointing to the best expected time to treat esophageal cancer cells with cisplatin (Figure6G).

  1. There are many carcinogenic cell lines, and they may differ functionally, structurally and genetically. Are you suggesting that this relationship between the Per2 expression and chemotherapy is universal among all mammalian esophageal cancers?

We made three major observations in esophageal cancers:

  • Clock related genes are deregulated in esophageal cancer biopsies as compared to normal tissue and tumor progression does not lead to a significant repression of the clock related genes.
  • Although clock related genes are deregulated in esophageal cancer (eSCC) cells as compared to normal esophageal cells in human, PER2 keeps oscillating and sensitizes cells to chemotherapy.
  • The pattern of expression of clock related genes is not profoundly deregulated in eSCC cells as compared to normal esophageal cells in mouse, and Per2 keeps oscillating and sensitizes cells to chemotherapy.

Obviously, we cannot claim that the relationship between PER2 expression and chemotherapy is universal. Even by investigating this relationship in dozens of lines we could not draw this conclusion. Our results are in line with those recently published (Van der Watt et al., Mol. Cancer Res. 2020) and suggest that this expression of PER2 can oscillate in a daily manner in esophageal cancer cells. By showing the relationship between PER2 expression and sensitivity to chemotherapy in two different species, we rather suggest that the mechanisms involved are conserved and not universal. Still, it would be interesting to try to decipher the molecular bases of this relationship, especially in vivo, by using more cell lines as well as patient derived xenografts (PDX). We have included these perspectives in the revised version of our manuscript (Lines 581-583).

  1. You have the data for 24h and 36h after dexamethasone application. What about 48h and 60h? There needs to be at least two cycles of the oscillation.

We would like to thank the Reviewer#3 for this excellent suggestion. We have repeated the experiments of Annexin V staining to measure apoptosis 48 hours after the end of the cisplatin treatment (Figure6D). We treated eSCC cells 24, 36, 48 and 60 hours after synchronization as suggested by the Reviewer#3. Interestingly, we found that cells are more sensitive to cisplatin-induced apoptosis 60 h (High Per2 expression) as compared to 48 h after synchronization (Low Per2 expression) (Figure6E-F). Hence, these new results clearly show that the Per2 expression level really correlates with the sensitivity to cisplatin. These new data have been included in the revised version of our manuscript (Figure6).

Round 2

Reviewer 2 Report

None

Reviewer 3 Report

The revision is satisfied to publish.